# Prospect Theory and the Favorite Long-Shot Bias in Baseball

**James Nutaro**

Computational Sciences and Engineering Division, Oak Ridge National Laboratory, Oak Ridge, TN 37831, USA; nutarojj@ornl.gov

**Abstract:** We provide new evidence of a favorite long-shot bias for bets placed on baseball games. Our analysis uses the difference of mean run differentials as an observable proxy for the probability of a team to win. When baseball is viewed through this proxy, we see that bettors believe favorites are less likely to win than they actually are and long-shots more likely. This result is consistent with prospect theory, which suggests that large and small probabilities are poorly estimated when making decisions with risk.

**Keywords:** sports betting; prospect theory; favorite long-shot bias

## 1. Introduction

We provide a new analysis of the favorite long-shot bias in baseball motivated by the possibility of prospect theory as an explanation (Ottaviani and Sørensen 2008).[1] Specifically, we hypothesize that an observable metric offers an approximate measure of probability, and that bets placed in relation to this metric assign a probability of winning to strong contenders that is less than its true value. In other words, there is a disinclination to bet on favorites and a preference for long-shots. The proposed metric appears to offer a visible estimate of true probabilities, and so fits neatly into the model of perception that is central to prospect theory.

Prospect theory was proposed to explain deviations of observed behavior from what is predicted by expected utility theory, which has been prevalent in classical economics (Levy 1992). In the context of decision making under uncertainty, such as when placing a wager, expected utility theory posits that the bettor seeks a maximum expected return. If the expected return on both sides of a bet are the same, then there should be no systematic bias towards favorites or long-shots. A rational actor that chooses to make a wager is as likely to bet on one side as the other.

This is not observed in practice. Rather, when given good estimates of true probabilities, an actor will tend to act as though low probabilities were more likely than the data justify, and the reverse for high probabilities (Levy 1992). We can imagine the actor viewing probabilities through a distorting lens that magnifies small probabilities and diminishes large ones. If we assume the value of money at stake is not similarly distorted (a simplification we make here that is not intrinsic to prospect theory), then bettors looking through this lens will be inclined to bet on a likely loser, the long-shot, and disinclined to bet on a likely winner, the favorite.

We present evidence for the existence of this bias in bets placed on baseball games using data for the 2010–2022 seasons, a total of 30,510 games. The analysis suggests that a favorite long-shot bias is apparent at extreme values of our observable metric. However, no significant bias appears when games are away from these extremes. This finding is consistent with prospect theory, wherein high probabilities are discounted, low probabilities are overstated, and middling probabilities are not subject to significant distortion (Kahneman and Tversky 1979).

Woodland and Woodland (1994, 2003) have argued in favor of a reverse favorite long-shot bias in baseball, though the significance of these findings have been questioned by

Gandar et al. (2002). A reverse favorite long-shot causes bettors to assume a favorite is more likely to win than it actually is and the long-shot less likely. Our analysis lends weight to the position taken by Gandar et al. (2002) that baseball betting may be unremarkable in this respect, with bettors favoring long-shots as they do in many other sports (Newall and Cortis 2021).

Yu et al. (2022) construct a model that can be parameterized to produce either a favorite–long-shot bias or its reverse. The ability of this model to fit historical data for baseball is demonstrated using the same data that Woodland and Woodland used to argue for a reverse bias. Curiously, the model appears to be capable of producing both the favorite–long-shot bias and its reverse for those data. Of course, the model embodies several theoretical effects apart from a distorted perception of probability, thereby exhibiting greater flexibility when fitting the data. Their model offers intriguing theoretical possibilities, but statistical uncertainty necessarily grows with the number of free parameters when the available data remain fixed. Given a paucity of data, our analysis favors simplicity and the accompanying clarity about statistical uncertainty.

Our method of analysis most closely resembles a former study of prospect theory in the context of horse racing by Snowberg and Wolfers (2010), in which distinct models are offered to distinguish bias produced by a preference for risk and bias produced by a distorted perception of probabilities. Snowberg and Wolfers conclude, as we do, in favor of distorted perception. Our approach differs chiefly in the method used. Their model uses offered odds and returns, omitting facts about the contestants. Our innovation is to introduce a proxy metric for probability that is specific to baseball and intuitively, if coarsely, observable by fans of the game. In this way, we offer a more direct test of biased perception by showing that offered odds deviate in the expected way from approximately observable probabilities.

The proposed model of decision making assumes that the relative strength of the contestants is a causal factor in how fans place bets on the game. We further assume that the offered odds are derived from how wagers are placed, rather than any insight of the bookmaker concerning the game. Hence, the offered odds are a statement of belief by the betting public concerning the probabilities of a win or loss, and this statement is made after observing the relative strengths of the teams.

Naturally, relative strength is expected to be a factor in the outcome of a game, and our data appear to support this conclusion. We find the proposed model to be in agreement with prospect theory in the sense that the perceived strengths of the teams are distorted when one team is particularly strong in relation to the other.

## 2. Bookmaker Odds and Efficient Markets

Whenever possible, the bookmaker offers odds that make a profit regardless of the outcome of a game. If a fraction $w$ of wagers are placed on team $A$ and $1 - w$ on team $B$, then the bookmaker wants the payout on team $A$ to be less than $1 - w$; that is, the payout to be less than the money kept because $B$ loses. For the same reason, the total payout on team $B$ should be less than $w$.

In an efficient market, all information is accounted for by the betting public and wagers are placed in proportion to true probabilities; $w$ will closely approximate the true probability of a win by $A$ and likewise $1 - w$ in relation to $B$. If we assume an efficient market and a bookmaker that sets odds according to the probability implied by $w$, then a bettor's expected utility for a wager on either side will be less than zero.

American odds are stated as a money line, which is translatable to the implied probability of a win and the payout on a bet. The anticipated winner has a money line set at $-100$ or less. If $A$ is the expected winner and the money line for $A$ is $-m_A$ then

$$w = -m_A / (-m_A + 100) \qquad (1)$$

is the implied probability of a win for $A$. The monetary gain on a dollar bet is

$$g_A = 100/-m_A \,. \tag{2}$$

The anticipated loser has a money line set at 100 or more. A money line $m_B$ for $B$ translates to an implied probability

$$1 - w = 100/(m_B + 100) \tag{3}$$

and the monetary gain on a dollar bet is

$$g_B = m_B/100 \,. \tag{4}$$

If $w$ is close to the true probability of a win, the expected utility of a dollar bet on $A$ is

$$
\begin{aligned}
wg_A - (1 - w) &= \frac{-m_A}{-m_A + 100}\frac{100}{-m_A} - \left(1 + \frac{m_A}{-m_A + 100}\right) \\
&= \frac{-m_A + 100}{-m_A + 100} - 1 \\
&= 0 \,.
\end{aligned} \tag{5}
$$

The expected utility of a dollar bet on $B$ is

$$
\begin{aligned}
(1 - w)g_B - w &= \frac{100}{m_B + 100}\frac{m_B}{100} - \left(1 - \frac{100}{m_B + 100}\right) \\
&= \frac{m_B + 100}{m_B + 100} - 1 \\
&= 0 \,.
\end{aligned} \tag{6}
$$

Of course, the bookmaker selects $-m_A$ and $m_B$ such that these utilities are slightly negative, thereby assuring a loss for the bettor and profit for the bookmaker if the market sets $w$ efficiently.

However, if the implied probability $w$ of a win for $A$ is less than the true probability $p$ (here we depart from the assumption that $w \simeq p$), then the betting public has an opportunity for profit. In this case, a bet on $B$ loses money for the bettor and a bet on $A$ is profitable. The expected gain of a bet on $B$ is calculated by rearranging (3) and (4) to obtain

$$g_B = \frac{1}{1 - w} - 1$$

and substituting this into (6)

$$(1 - p)g_B - p = (1 - p)\left(\frac{1}{1 - w} - 1\right) - p = \frac{1 - p}{1 - w} - 1 \,.$$

If $w < p$ then $1 - w > 1 - p$ causing a loss for the bettor and profit for the bookmaker. In the same way, using (1), (2), and (5), we find that for a bet on $A$

$$pg_A - (1 - p) = p\left(\frac{1 - w}{w}\right) - (1 - p) = \frac{p}{w} - 1 \,.$$

In this case, the bookmaker loses and the betting public profits.

## 3. Conditions for Inefficiency

Prospect theory posits that the perceived value of a chancy proposition is not governed by a clear view of probabilities and gains. Instead, the perceived likelihood of an outcome is a function $u(p)$ of the true probability $p$. Similarly, the perceived value is a function $v(g)$

of the objective gain $g$, and a loss $\ell$ is likewise valued at $g(\ell)$. The net perceived value of the proposition is

$$u(p)v(g) - u(1-p)v(\ell) \,.$$

In examining the behavior of the betting public, we assume that the money line is set in response to the proportion of bets on either side. Specifically, the bookmaker assumes an efficient market and sets the money lines to reflect $w$ and $1 - w$. We take $v$ to be the identity because it is evident in the money lines, which reflect the perception of value by the betting public.

We are left with $u$ to explain observed behaviors that are distinct from an efficient market. If $p$ is the true likelihood of team $A$ winning the game, the prospective utility of a dollar bet on $A$ is

$$u(p)g_A - u(1-p)$$

and for team $B$ is

$$u(1-p)g_B - u(p) \,.$$

The quantities $u(p)$ and $u(1-p)$ are expressed in the wagers placed and the money lines set accordingly. For team $A$, the money line $-m_A$ implies a perceived probability

$$u(p) = -m_a/(-m_A + 100) \tag{7}$$

and for team $B$, $m_B$ implies

$$u(1-p) = 100/(m_B + 100) \,. \tag{8}$$

A deviation of $u$ from the identity will be detectable as a difference between the probability $p$ that is measurable from historic data and the implied probability $u(p)$ (or $u(1-p)$) in the historic money lines, as calculated with (7) (or (8)).

*A Difference of Mean Run Differentials*

Evidence supporting prospect theory comes from laboratory experiments in which probability $p$ and gain $g$ are shown to the experimental subject. In the laboratory there are two consistent features of the measured $u(p)$. First, when the probability of a win is suitably small, the perceived probability will be greater than the true probability. Second, when the probability of a win is suitably large, the perceived probability will be less than the true probability.

In a betting market, the bettor knows $g$ but does not know $p$. Therefore, the bettor must infer $p$ from observable factors that are believed to have predictive value. If the observations anticipated by prospect theory are applicable to a betting market, then $u$ must act on some observable factor that substitutes for probability in the thinking of the betting public.

The factor we examine is the difference in mean run differentials over a span of games. The proposed metric is similar to (indeed, inspired by) run differentials and the Pythagorean win–loss formula; see, e.g., the derivation of the latter by Miller (2007). However, we incorporate data from both teams to anticipate the outcome of a game, rather than data from a single team to anticipate the outcome of a season.

Of course, a bettor is unlikely to perform the calculation suggested here. Nonetheless, there is a general awareness of it fostered by the outcomes of games played by the teams involved in a coming match. A strong offence is indicated by runs scored; a strong defense by runs prevented. Injuries, trades, and other events ensure that a strong or weak performance in the past loses its influencing force over time. The mean run differential codifies this intuition in a form useful for calculations.

A team's mean run differential is the difference between total runs allowed and total runs scored, divided by the number of games considered. We look at the most recent $n$ games played by each team. The quantity $o_A$ is the total number of runs scored by $A$ in

those games, $o_B$ the total number of runs scored by $B$, $d_A$ the total runs scored against $A$, and $d_B$ the runs scored against $B$. The mean run differential $\Delta_{AB}$ at the start of a game between $A$ and $B$ is

$$\Delta_{AB} = \frac{o_A - d_A}{n} - \frac{o_B - d_B}{n} \ . \tag{9}$$

For example, consider the schedule of games shown in Table 1. For $n = 2$, we have $o_{\text{Cubs}} = 12$, $o_{\text{Nationals}} = 9$, $d_{\text{Cubs}} = 8$, $d_{\text{Nationals}} = 12$, and

$$\Delta_{\text{Cubs,Nationals}} = \frac{12 - 8}{2} - \frac{9 - 12}{2} = 1.5 + 2 = 3.5.$$

In a match between the Cubs and Nationals, the Cubs are expected to have an advantage.

Given $\Delta_{AB}$ preceding a game, the probability of a win is a function $f$ of $\Delta_{AB}$. Using $\Delta_{AB}$ in place of probability in our model of perception, the perceived probability becomes $u(f(\Delta_{AB}))$. A deviation of $u$ from the identity is reflected in the difference

$$u(f(\Delta_{AB})) - f(\Delta_{AB}) \ .$$

**Table 1.** A schedule of completed games.

| Month/Day | Visitor @ Home | Visitor Runs–Home Runs |
|---|---|---|
| 5/23 | Cubs @ Reds | 7–4 |
| 5/23 | Dodgers @ Nationals | 10–1 |
| 5/22 | Diamondbacks @ Cubs | 4–5 |
| 5/22 | Nationals @ Brewers | 8–2 |

## 4. Results

Our data consist of opening money lines and scores for all games in the 2010 through 2022 major league seasons; these were obtained from Sports Statistics.[2] A challenge encountered when calculating $u(f(\Delta_{AB}))$ and $f(\Delta_{AB})$ using these data is that any specific value of $\Delta_{AB}$ has a small representation. To mitigate this problem, we calculate mean values of $f$ and $u$ over sets of games with $\Delta_{AB}$ greater than a threshold $\delta$.

For each value of the mean run differential appearing in our data, we select a threshold $\delta$ equal to this value and then construct two sets using this threshold. The first set comprises games with $\Delta_{AB} > \delta$ and the second set comprises games with $\Delta_{AB} \leq \delta$. Then we calculate the averages and confidence intervals of

$$\{ f(\Delta_{AB}) \mid \Delta_{AB} > \delta \} \,,$$
$$\{ f(\Delta_{AB}) \mid \Delta_{AB} \leq \delta \} \,,$$

and likewise for $u$.

We calculate $\Delta_{AB}$ over the previous ten, twenty, forty, and sixty games played by each team in our data. This choice corresponds to major league baseball's ten- and sixty-day injured lists with two points in between, but it is otherwise an arbitrary selection. However, examinations of other choices did not materially alter our conclusions.

All quantities calculated are assumed to be drawn from a normally distributed random variable. Plots show the mean as a solid line and the 95% confidence interval is a band about that line. The confidence interval is calculated using Wilson's method (Agresti and Coull 1998; Brown et al. 2001). We stop plotting when the 95% confidence interval exceeds a width of 10%.

The implied probability $u$ is calculated from historical money lines and is shown in green. The probability of a win $f$, shown in purple, is calculated from historical wins and losses. The difference $u - f$ between the implied and true probability is in red, and its scale is the second $y$ axis.

Plots of $f$ and $u$ for $\Delta_{AB} > \delta$ are shown in Figures 1 and 2. When the mean run differential is sufficiently small, it is calculated over essentially all of the games. In this case,

the implied probability is consistently below the actual probability. This is indicative of the money lines being close to the true probability and the bookmaker offering odds to ensure a profit.

The set of games is winnowed as we move to the right. For these increasingly refined subsets the implied probability becomes, at least, indistinguishable from the true probability. If we look at the difference of the means there is a clear trend. The implied probability indicated by the wagers approaches and may becomes less than the true probability. The betting public believes the chance of a win is smaller than it actually is. This is in agreement with prospect theory.

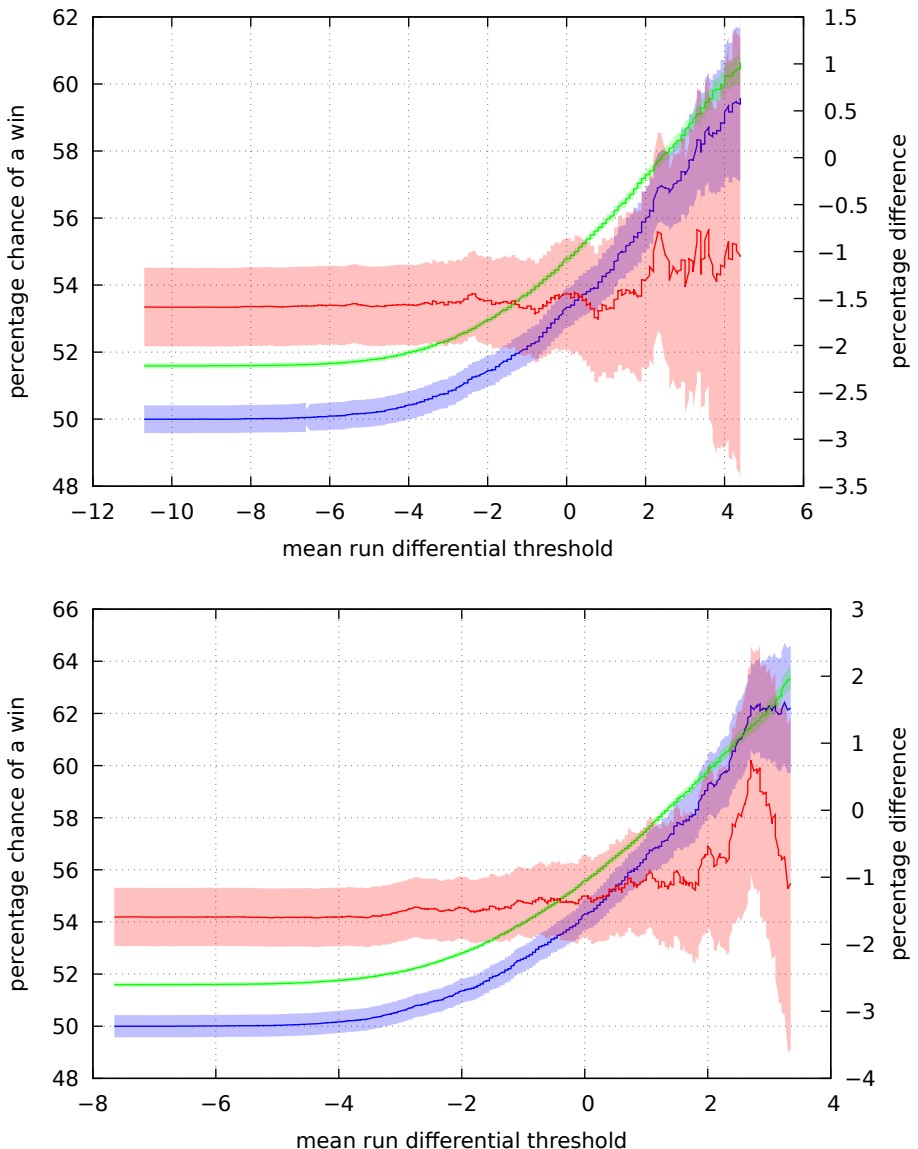

**Figure 1.** Implied probability (green), true probability (purple), and the difference (red) as the mean run differential increases. These plots are calculated using ten-day (**above**) and twenty-day (**below**) means. The colored bands show 95% confidence intervals.

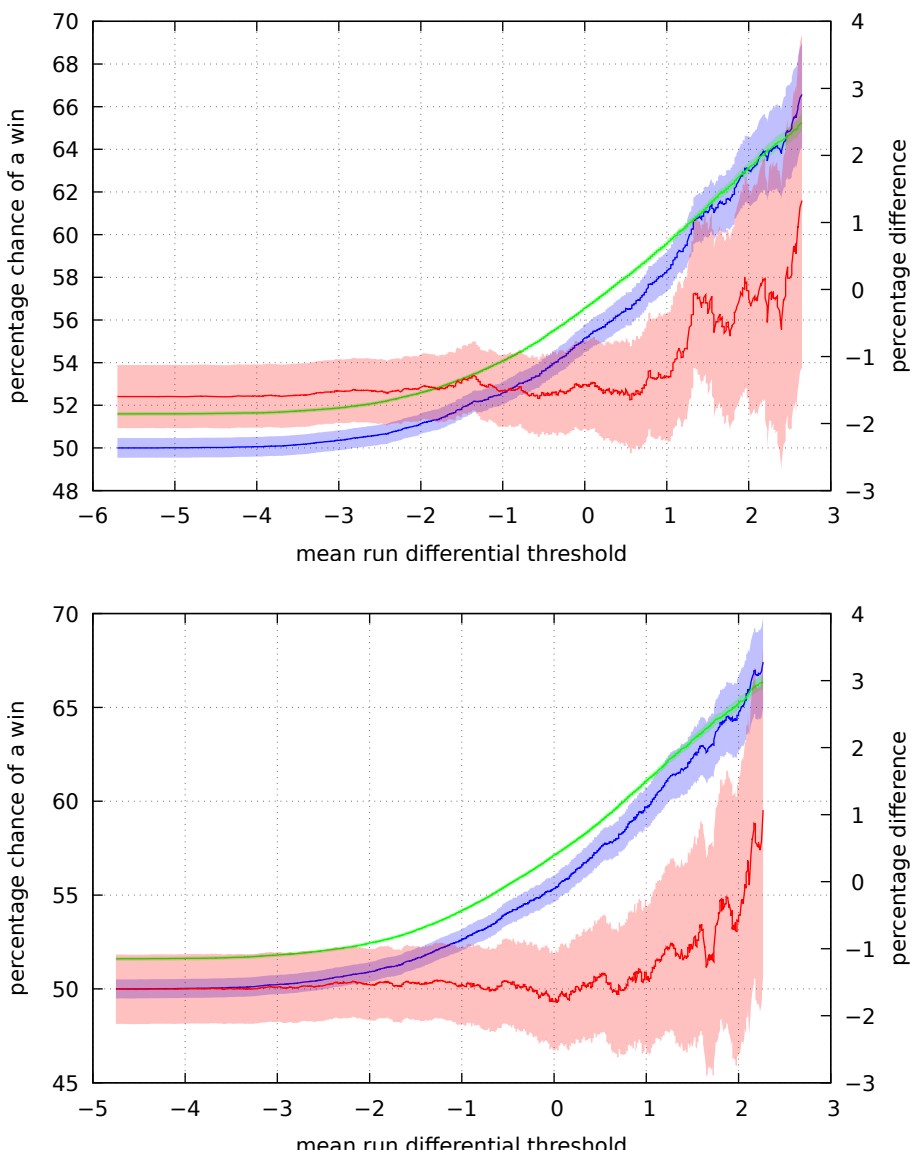

**Figure 2.** Implied probability (green), true probability (purple), and the difference (red) as the mean run differential increases. These plots are calculated using forty-day (**above**) and sixty-day (**below**) means. The colored bands show 95% confidence intervals.

In Figures 3 and 4 we plot $f$ and $u$ for $\Delta_{AB} \leq \delta$. The plots begin with the implied odds 1% below the true odds, which is consistent with the betting public having a good estimate of the true odds and the bookmaker setting the money line to ensure a profit. In this case, the mean is calculated using essentially all of the games.

The set of games becomes increasingly refined as we move left. Moving far enough to the left, we see that for a sufficiently negative mean run differential, the implied probability of a win diverges from the true probability. Examining the difference of $u$ and $f$, the implied probability becomes greater than the the bookmaker's margin of 1%. As anticipated by prospect theory, the betting public believes a win is more likely than it actually is.

While bias is observable over each of the intervals examined, the effect is most pronounced for the forty- and sixty-day periods. Over the ten- and twenty-day periods, the trend towards bias weakens or reverses itself for sufficiently extreme values of $\Delta_{AB}$. In part, this may be due to greater variety in a team's performance when viewed over short time scales. Strong teams suffer losing streaks and weak teams enjoy strings of wins, and these

obfuscate the relation between the perceived and actual chances of a win. In these cases, bettors may focus on near term trends while discounting a general superiority of play.

A consistent trend is seen in the forty- and sixty-day means. By taking the mean of a larger set of games, volatility in play and any near term bias will tend to balance. Winning streaks will cancel the effects of losing streaks and vice versa over a sufficiently large period of time. When near term effects are smoothed out in this way, the underlying preference for long-shots over favorites becomes apparent.

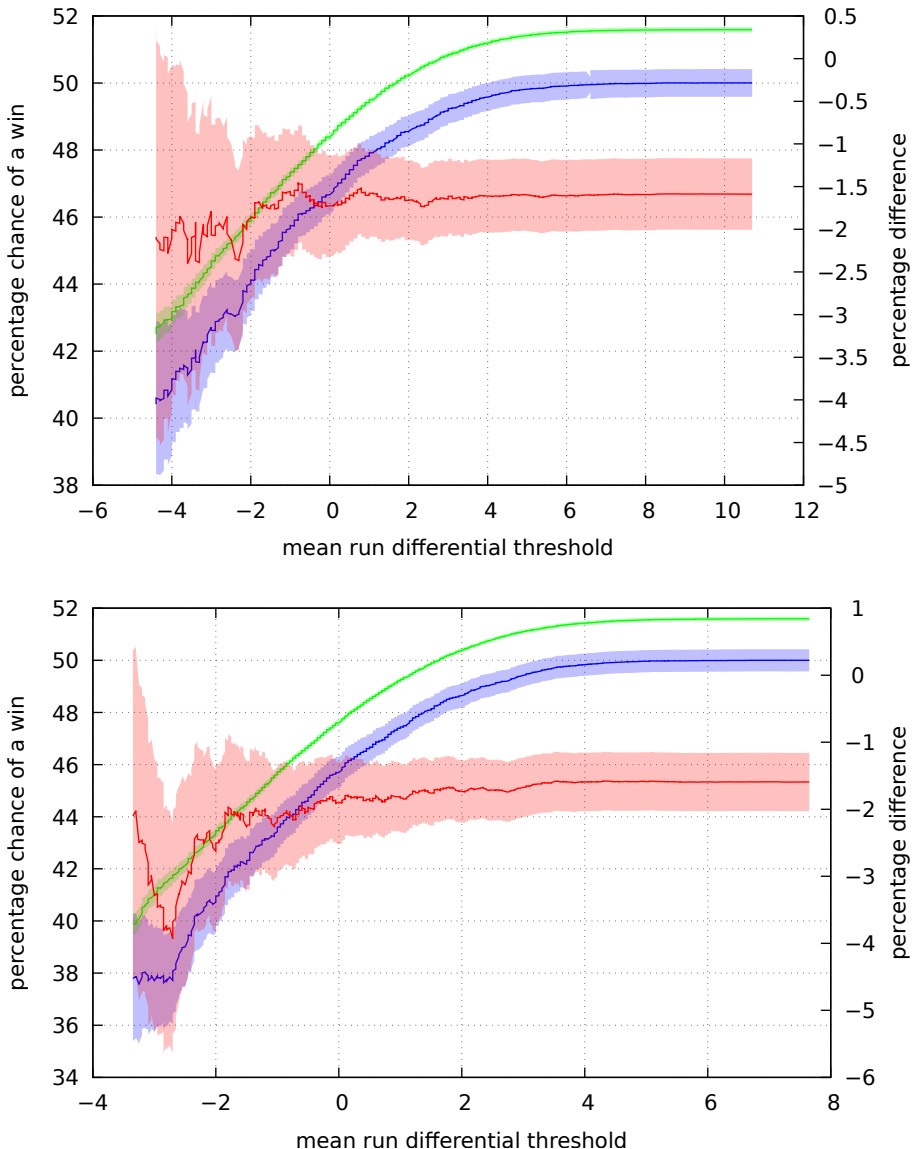

**Figure 3.** Implied probability (green), true probability (purple), and the difference (red) as the mean run differential decreases. These plots are calculated using ten-day (**above**) and twenty-day (**below**) means. The colored bands show 95% confidence intervals.

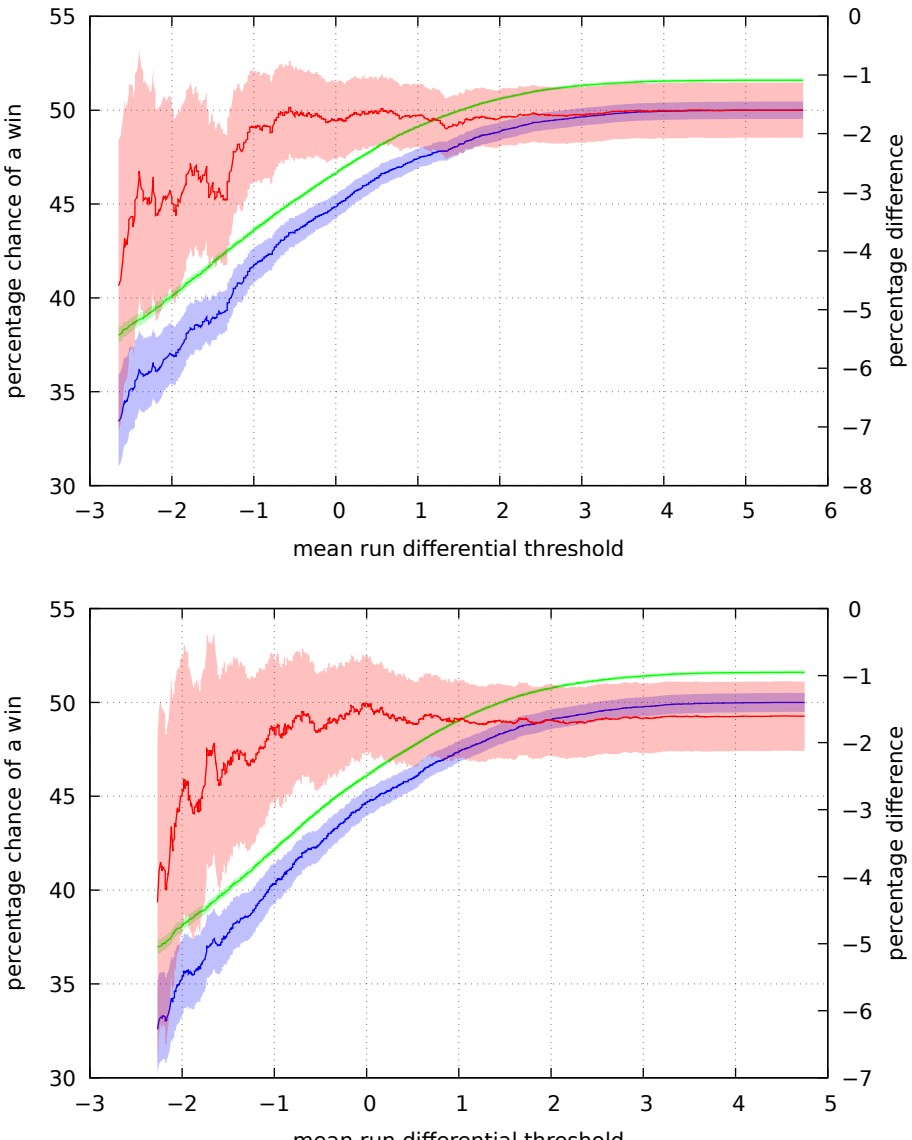

**Figure 4.** Implied probability (green), true probability (purple), and the difference (red) as the mean run differential decreases. These plots are calculated using forty-day (**above**) and sixty-day (**below**) means. The colored bands show 95% confidence intervals.

## 5. A Betting Strategy

A sketch of a possible betting strategy emerges when we look at a larger data set. Figure 5 plots the percentage chance of a win when the difference of mean run differentials exceeds a given threshold (shown on the *y*-axis). Win percentages are calculated using data from the 1901–2022 Major League Baseball seasons provided by Retrosheets.[3] Each line terminates when the 95% confidence interval exceeds $\pm 1\%$.

Percentages on the *x*-axis are expressed as a money line, which is calculated using (1). Unfortunately, money line data are not available for this period. Because $n = 10$ does not appear to produce the prospect theory effect (see Figure 1) and $n = 60$ does not produce sufficient observations for the desired confidence interval, we use $n = 20, 30, 40,$ and $50$ to illuminate the relationship between mean run differential and the money line.

Using this plot, pick a number $n$ (how is an open question) and calculate the mean run differential for your team using (9). Find the point on the plot corresponding to this calculated value and the offered money line. If the point is to the right of the curve for $n$ then the offered odds are a good bet. In Figures 1 and 2 the implied and true probabilities

intersect at a mean run differential of approximately two. We can conclude that profitable betting opportunities will tend to occur in the upper half of Figure 5.

However, this strategy makes several unproven assumptions. Most importantly, it assumes that the relationship between wining and mean run differential is essentially static in time and that mean run differential is sufficient to forecast team performance (i.e., there are no important hidden factors). Whether the strategy can be profitable in practice remains to be seen.

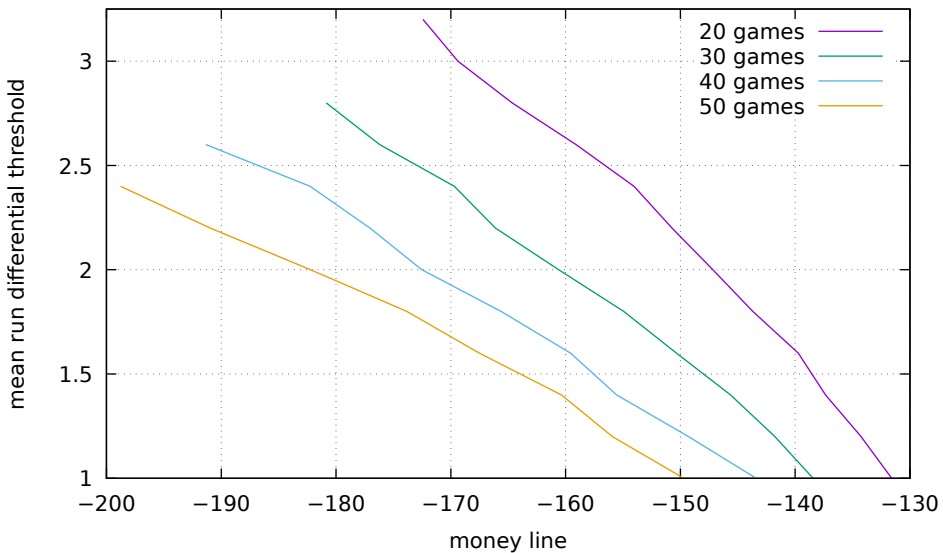

**Figure 5.** Probability of a win, expressed as a money line, when the difference of mean run differentials exceeds a given threshold.

## 6. Conclusions

Our results suggest that probabilities assigned to outcomes by baseball fans, as reflected in the offered odds, underweight the prospects of a strong team and overweight those of a weak team. This is consistent with experimental evidence supporting prospect theory, in which small and large probabilities are poorly estimated. Consistent with intuition, our analysis also suggests that past performance on the playing field is a strong indicator of the actual and perceived probability of a win.

A secondary outcome of our analysis is to suggest a definite market inefficiency when very strong teams face very weak teams, and that this inefficiency is best exploited by a taking a long view. A bias for recent events has been long suspected in sports betting. At least two other former studies of sports betting have come to the same conclusion.

Krieger et al. (2021) have suggested that a bias for recent events may offer profitable opportunities for gamblers who take the long view. For over/under bets, which are distinct from the win/loss bets we examine, Woodland and Woodland (2016) have argued that market inefficiencies create profitable strategies for betting on baseball games. There also, a long view is inseparable from the advantage.

A consequence of our findings would appear to be an opportunity for profitable bets. However, it is prudent to question the adequacy of a single metric when calculating the outcomes of a game. For instance, the difference of mean run differentials omits any explicit consideration of past wins and losses.

To illustrate a possible consequence of this omission, suppose the Cubs win twenty games by a single run each for a mean run differential of one. The Nationals have two wins at 19-0 and eighteen losses at 0-1 and so also have a mean run differential of one. The difference is zero and the Cubs and Nationals would appear to be evenly matched, but the erratic performance of the Nationals raises doubts. Similar questions can be raised about the possible influences of pitchers and fielders, which team is home and which away, and a host of other factors.

However, our analysis also suggests that if such possibilities exist for win/loss bets, then they are likely to be rare. The infrequency of profitable betting opportunities goes some way towards reconciling two apparently contradictory positions: a persistent bias that should lead to bookmaker losses and the continuing existence of profitable sports books.

**Funding:** This research received no external funding.

**Data Availability Statement:** Data available in a publicly accessible repository that does not issue DOIs Publicly available datasets were analyzed in this study. This data can be found here: https://oldtimebaseball.net/public/risks-article-data.zip (accessed on 4 May 2023).

**Acknowledgments:** This manuscript has been authored by UT-Battelle, LLC under Contract No. DE-AC05-00OR22725 with the U.S. Department of Energy. The publisher, by accepting the article for publication, acknowledges that the U.S. Government retains a non-exclusive, paid up, irrevocable, world-wide license to publish or reproduce the published form of the manuscript, or allow others to do so, for U.S. Government purposes. The DOE will provide public access to these results in accordance with the DOE Public Access Plan (http://energy.gov/downloads/doe-public-access-plan) (accessed on 4 May 2023).

**Conflicts of Interest:** The author declares no conflict of interest.

## Notes

[1]    Referred to there as a misestimation of probabilities.
[2]    https://sports-statistics.com/sports-data/mlb-historical-odds-scores-datasets/, accessed on 4 May 2023.
[3]    https://www.retrosheet.org/, accessed on 4 May 2023.

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
