# Peer review of "Prospect Theory and the Favorite Long-Shot Bias in Baseball"

_risks, doi:10.3390/risks11050095_

Round 1
Reviewer 1 Report
There's a chance I've gotten myself turned around, but I feel as if the 'overvalued' and 'undervalued' comments in the abstract are flip-flopped. i.e., aren't bettors UNDERVALUING favorites? (They're not willing to bet on them at the rates/prices they should). And aren't bettors OVERVALUING underdogs (They're willing to beg on them on rates/prices that they should not).
I might make a (fun) note that, (as you allude to analysis like 'Pythagorean' run differential for baseball has become more common) run differentials did not use to be part of a traditional 'standings' page of MLB...I don't think I saw it growing up...and now it's rather 'front and center.'
I think there's a typo in the calculations of the example re: Cubs/Nationals...isn't the Nationals value -1.5, rather than -2, because they've scored 9 runs combined, not 8 (and this seemed correctly noted in the corresponding text).
Here are my bigger 'concerns' or wish list...and the common thread is the desire for more easy-to-visualize concrete examples/illustrations. (The mathematical exposition herein is nice and on point)...
As a reader (and a fan/bettor!) I want to sink my teeth into WHAT KINDS of 'delta' actually precipitate mispricing. I can see the exposition (for different lengths of backward-looking inputs) of the prospect theory/favorite longshot stuff well in a mathematical sense...
BUT, is a -175/+155 baseball game BIG ENOUGH of an ex ante mismatch so that we might say it's likely to be mispriced, all else equal? Can we say that is the case with statistical significance?
Or do I need a -250/+230 baseball game to see the prospect theory kick in? When it does 'kick in' is it enough for bettors to profit and overcome the commission?
One potential way to answer this, and maybe all at once...I'm envisioning a table that just shows me the 'winning percentage' and the PROFIT if I systematically bet $100 on all teams at -400 or better...then -390...then -380...ALL the way down to +400, e.g. A big challenge to that is where do you draw the lines for the 'bins'? It's unlikely you have enough data of exactly -390 favorites (that extreme might not even exist in MLB, exactly)...But is -350 to -390 too wide? Too narrow? I'd just base it on how my data looks on the whole...
I think that'd be a very useful illustration. But I'm biased by two things: One, I'm primarily an empiricist, so I'm wired to want to see those things and think they're a valuable use of time and white space in an article. Some outlets don't care as much about that, and the mathematics are more important.
And the second matter I basically already alluded to...but it's SPACE/WORD COUNT. Such an illustration could pinch and be infeasible based on the requirements the authors are facing. In that situation, I'd STILL rather see such a table then some of the exposition and figures in the current draft, but I again admit that's likely a function of my own taste.
Author Response
There's a chance I've gotten myself turned around, but I feel as if the 'overvalued' and 'undervalued' comments in the abstract are flip-flopped. i.e., aren't bettors UNDERVALUING favorites? (They're not willing to bet on them at the rates/prices they should). And aren't bettors OVERVALUING underdogs (They're willing to beg on them on rates/prices that they should not).
Yes, you are correct. In the end, I removed that sentence from the abstract. I found the language confusing and felt it was better to just stick with a straightforward statement about perceived and actual probabilities.
I might make a (fun) note that, (as you allude to analysis like 'Pythagorean' run differential for baseball has become more common) run differentials did not use to be part of a traditional 'standings' page of MLB...I don't think I saw it growing up...and now it's rather 'front and center.'
A good observation. It also made we wonder why I didn’t mention the run differential directly in that paragraph, and so I have added that to the Pythagorean formulae as a predecessor.
I think there's a typo in the calculations of the example re: Cubs/Nationals...isn't the Nationals value -1.5, rather than -2, because they've scored 9 runs combined, not 8 (and this seemed correctly noted in the corresponding text).
Yes, you are correct. I’ve fixed the error.
Here are my bigger 'concerns' or wish list...and the common thread is the desire for more easy-to-visualize concrete examples/illustrations. (The mathematical exposition herein is nice and on point)...
As a reader (and a fan/bettor!) I want to sink my teeth into WHAT KINDS of 'delta' actually precipitate mispricing. I can see the exposition (for different lengths of backward-looking inputs) of the prospect theory/favorite longshot stuff well in a mathematical sense...
BUT, is a -175/+155 baseball game BIG ENOUGH of an ex ante mismatch so that we might say it's likely to be mispriced, all else equal? Can we say that is the case with statistical significance?
Or do I need a -250/+230 baseball game to see the prospect theory kick in? When it does 'kick in' is it enough for bettors to profit and overcome the commission?
One potential way to answer this, and maybe all at once...I'm envisioning a table that just shows me the 'winning percentage' and the PROFIT if I systematically bet $100 on all teams at -400 or better...then -390...then -380...ALL the way down to +400, e.g. A big challenge to that is where do you draw the lines for the 'bins'? It's unlikely you have enough data of exactly -390 favorites (that extreme might not even exist in MLB, exactly)...But is -350 to -390 too wide? Too narrow? I'd just base it on how my data looks on the whole...
I think that'd be a very useful illustration. But I'm biased by two things: One, I'm primarily an empiricist, so I'm wired to want to see those things and think they're a valuable use of time and white space in an article. Some outlets don't care as much about that, and the mathematics are more important.
I have attempted to provide this information by adding a plot that shows the mean run differential as a function of probability expressed as a money line. If a combination of mean run differential and money line (i.e., a point on the plane) is to the right of the curve, then the game is a good bet. The key point emphasized by the plot (and accompanying text) is that both data points are needed to decide if a misperception has occurred and a good betting opportunity produced. The thresholds (either money line or mean run differential) at which the prospect theory effect becomes apparent can also be estimated from the new plot.
See Section 5: A betting strategy.
And the second matter I basically already alluded to...but it's SPACE/WORD COUNT. Such an illustration could pinch and be infeasible based on the requirements the authors are facing. In that situation, I'd STILL rather see such a table then some of the exposition and figures in the current draft, but I again admit that's likely a function of my own taste.
Fortunately, I am below the suggested minimum 4,000 word count and have plenty of room for the additional material.
Reviewer 2 Report
see the attached file

Author Response
This paper introduces a novel metric, referred to as Mean Run Differential (MRD), for assessing the relative strength of two baseball teams in a match. The metric is based on historical matches and involves dividing them into upper and lower level sets with increasing MRD thresholds. Empirical analyses are conducted on these MRD-based level sets. The paper proposes that there is a tendency to underestimate the probabilities assigned to the favorite team, which is consistent with prospect theory. However, this bias appears to weaken or reverse for matches with extreme MRD disparities.
Overall, I find this paper to be both interesting and well-grounded. The methodology employed is rigorous, and the proposed MRD metric has the potential to be useful in a broader context. To further enhance the paper, the author may want to consider:
- The author may want to double-check the equations between line 68 and line 69 for accuracy and clarity.
There was an error in the equation for Gb and for the following equation. This implies (as one should expect) that a bet on A (the favorite) makes money and a bet on B (the long-shot) loses money. I have corrected the error and text to reflect the correction. Thank you for the careful review!
- Since prospect theory is a central theme of this paper, it would be beneficial to provide a more detailed introduction or discussion about the theory in either Section 1 Introduction or Section 3 Conditions for Inefficiency.
The introduction was extended to include an informal review of the elements of prospect theory that are relevant to the analysis presented in the paper. Please see the revised first half of Section 1.
- It would be useful to consider differentiating between wins and losses when aggregating runs scored and lost. For instance, what if Team A won nine games with a score of 5-4 but lost one game with a score of 1-10, while Team B won one game with a score of 10-1 but lost nine games with a score of 4-5? In this scenario, the 1 MRD is zero, but this might not accurately reflect the teams’ relative strengths. Therefore, the author may want to add a discussion about this aspect, assuming that the team with the most runs scored wins the game.
This is an excellent suggestion for future work and highlights one of the many factors that are absent from the model. A brief discussion of this and related possibilities has been added to the conclusions section.
- The paper could benefit from a review of related literature, such as [SW10] and [YGW22], to provide a comparison of this paper’s approach with others in the field. This would help to identify what differentiates this paper from previous works.
References
[SW10] E. Snowberg and J. Wolfers. Explaining the favorite–long shot bias: Is it risk-love or misperceptions? Journal of Political Economy, 118(4):723–746, 2010.
[YGW22] D. Yu, J. J. Gao, and T. Y. Wang. Betting market equilibrium with heterogeneous beliefs: A prospect theory-based model. European Journal of Operational Research, 2022
The discussion of [SW10] has been extended somewhat in the introduction to more clearly explain how our method of analysis differs (arguably, we offer a more direct test for misperception) while coming to the same conclusion (that misperception is an explanatory cause for long shot - favorite bias).
A discussion of [YGW22] has been added to the introduction. We note that our method of analysis is necessarily less flexible than the model proposed there, but what is lost in theoretical sophistication is gained in the ability to make statements about statistical certainty.
Reviewer 3 Report
This paper provides new evidence of a favorite long-shot bias for bets placed on baseball games. The analysis uses the difference of mean run differential as an observable proxy for the probability of a team to win. The result is consistent with prospect theory, which suggests that large and small probabilities are poorly estimated when making decisions under risk. The following suggestions are given to improve the quality of the paper:
1. The experiments can be fruited and improved to prove the results.
2. More mathematical expressions and formulas can be given.
3. More information about prospect theory can be added.
4. The introduction can be improved.
Author Response
This paper provides new evidence of a favorite long-shot bias for bets placed on baseball games. The analysis uses the difference of mean run differential as an observable proxy for the probability of a team to win. The result is consistent with prospect theory, which suggests that large and small probabilities are poorly estimated when making decisions under risk. The following suggestions are given to improve the quality of the paper:
- The experiments can be fruited and improved to prove the results.
We have extended the statistical analysis to look at the relationship between difference of mean run differentials and the probability of a win using data from 1901 - 1922. Unfortunately, money line data for that period is not available. See the new Section 5: A betting strategy.
- More mathematical expressions and formulas can be given.
That you for the suggestion. While I agree that a more detailed theoretical analysis would be interesting (e.g., as in Ref. [14]), I feel that this would be out of scope for a paper focused primarily on the analysis of historical data.
- More information about prospect theory can be added.
The introduction was extended to include an informal review of the elements of prospect theory that are relevant to the analysis presented in the paper. Please see the revised first half of Section 1.
- The introduction can be improved.
The introduction has been extended to include a brief review of the relevant concepts from prospect theory and to more precisely place our analysis in the context of related work.